# Mapping hippocampal-cerebellar functional connectivity across the human adult lifespan
Kavishini Apasamy [1], Samuel C. Berry [1], Marie-Lucie Read [2], Narender Ramnani[1,3] & Carl J. Hodgetts [1,3] ✉

The hippocampus and cerebellum are traditionally considered to support distinct memory systems, yet evidence from nonhuman species indicates a close relationship during spatial-mnemonic behaviour, with hippocampal projections to and from several cerebellar regions. However, little is known about this relationship in humans. To address this, we applied seed-based functional connectivity analysis to resting-state fMRI data from 479 cognitively normal participants aged 18–88 years. We identified significant functional correlations between the hippocampus and widespread areas of cerebellar cortex, particularly lobules HIV, HV, HVI, HVIIA (Crus I and II), HIX, and HX. Moreover, anterior hippocampus showed stronger connectivity with right Crus II, whereas posterior hippocampus was strongly connected to vermal lobule V. Finally, we observed age-related reductions in functional connectivity between the hippocampus and lobules HVI and HV. These findings provide insight into the topography of hippocampal-cerebellar functional organisation in humans and the influence of ageing on this system.

The ability to represent and navigate spatial environments is thought to be supported by an extended posteromedial navigation system in the brain that includes the hippocampus, as well as entorhinal, retrosplenial and para-hippocampal cortices[1–3] (see also ref. [4]). Critically, this brain system is also thought to undergo pronounced structural and functional alterations across the adult lifespan[5], acting as a 'hotspot' for age-associated neurodegenerative processes, such as tau and amyloid-beta accumulation[6,7]. In this context, it is important to better understand the organisation of the brain's navigation system, particularly by considering the influence of other cortical and subcortical areas.

The cerebellum, for instance, is classically associated with motor processing[8–10] but has been shown to be strongly connected with higher-order cognitive areas, such as prefrontal cortex[11–13]. In line with this, evidence from nonhuman species suggests that the cerebellum has functional and structural interactions with the hippocampus[14,15]. For example, electrophysiological studies in transgenic mice have shown that disrupting cerebellar cells alters hippocampal place cell firing and impairs navigational performance[16,17]. Similarly, optogenetic excitation or inhibition of midline and lateral cerebellar neurons has been shown to reduce the duration of left hippocampal seizure activity[18], and combined optogenetic stimulation-fMRI revealed that cerebellar stimulation (between lobule V and VI) increased dorsal hippocampal blood-oxygen level dependent (BOLD)

signal[19]. Taken together, these studies establish an argument that the cerebellum modulates the physiology of the hippocampus, as well as influencing hippocampal-dependent behaviour (e.g., spatial learning and memory).

While early electrophysiological evidence in primates and cats suggested a direct structural monosynaptic connection between these structures (as evidenced by short observed latencies between cerebellar fastigial nucleus stimulation and evoked hippocampal responses)[20,21], recent anatomical tracing work in rodents indicates connectivity through multisynaptic pathways[22]. Specifically, left hippocampus has been shown to receive input from distributed areas of cerebellar cortex, including bilateral and vermal lobule VI, lobule VIIA (Crus I) and paraflocculus via at least two relay stations[22]. This study also observed phase-locked theta coherence between cerebellar Purkinje cells in lobule VI, VIIA (Crus I) and the dorsal hippocampus during exploratory behaviour. Notably, evidence in macaque monkeys suggests that lobule VI and VIIA sends its output to the cerebellar fastigial nuclei[23] — the same area found by the above electrophysiological studies to evoke responses in the hippocampus. This suggests that topographically distinct areas of the cerebellum send their input to the hippocampus via distinct pathways.

Despite strong evidence of connectivity in nonhuman species, this is lacking in the human brain. Such evidence has the potential to extend and inform current neurobiological accounts of human memory and navigation,

[1]Department of Psychology, Royal Holloway, University of London, Surrey, UK. [2]Cardiff University Brain Research Imaging Centre (CUBRIC), School of Psychology, Cardiff University, Cardiff, UK. [3]These authors contributed equally: Narender Ramnani, Carl J. Hodgetts. ✉e-mail: carl.hodgetts@rhul.ac.uk

incorporating the cerebellum as a key structure, and may have implications for understanding memory decline in ageing and neurodegeneration.

Studies of healthy ageing in humans, for example, indicate that the cerebellum and hippocampus are both vulnerable to age-related structural changes[24–26] and exhibit comparable grey matter loss[27–29]. Notably, age-related reductions in cerebellar volume have been linked to poorer performance on hippocampal-dependent navigation tasks[30], suggesting that structural changes may disrupt hippocampal-cerebellar communication. Indeed, recent work in humans has found that age-related cerebellar atrophy is mainly localised to lobule HVI and lobule HVIIA (Crus I)[26]. Notably, these are two regions that have been shown to connect with the hippocampus in nonhuman animal studies[22], and to co-activate with the hippocampus during navigational learning in young human adults[31]. Despite this evidence, there remains a significant gap in understanding hippocampal-cerebellar connectivity in humans, and how it may differ across the lifespan.

Here, we aimed to examine these questions using resting-state functional MRI data from the Cambridge Centre for Ageing and Neuroscience (CamCAN) dataset[32]. We first sought to map patterns of bilateral hippocampal connectivity (independent of age) within the cerebellum. We hypothesised that the hippocampus would show a strong functional correlation with several areas of the cerebellum, including lobule VI and HVIIA (Crus I), as predicted by prior nonhuman animal tracing work[22] and electrophysiological evidence showing that stimulating fastigial nucleus (which receives input from these areas) leads to evoked responses in the hippocampus[23]. We also contrasted the connectivity of the left and right hippocampus, drawing on evidence that hippocampally-dependent tasks might be lateralised in the cerebellum[33]. It is also possible that a distinctive pattern of connectivity exists for each hemisphere—a question that remains unresolved as previous anatomical studies have primarily focused on left hippocampal connectivity[18,22]. The hippocampus is also thought to display functional gradients along its longitudinal (i.e., anterior-posterior) axis, arising in part from variation in connectional anatomy[34–36]. For instance, the anterior hippocampus preferentially connects to the amygdala and prefrontal cortex[37,38], whereas posterior hippocampus preferentially connects with the parahippocampal cortex[34]. As regions within these anterior and posterior hippocampal networks might differentially connect with cerebellar cortex (see e.g., prefrontal connections with lobule HVIIA[39]), we also predicted connectivity differences when contrasting anterior and posterior hippocampal seed regions. Finally, the large age range within the CamCAN dataset (18–87 years old) enabled us to examine how increasing age influences the degree and pattern of hippocampal-cerebellar functional connectivity. We predicted that the strongest age-related decreases in functional connectivity would be observed in lobules HVI and HVIIA, consistent with their vulnerability to age-related atrophy[26,28] and suggested involvement in spatial cognition[31].

We found widespread functional connectivity between the hippocampus and various areas of the cerebellar cortex, including lobules HIV-HVI, HVIIA (Crus I and II), HIX, and HX. The left and right hippocampus each showed preferential connectivity to a contralateral region of lobule HVIIA. The anterior hippocampus was most strongly connected with lobules HI-HV and bilateral regions of lobule HVIIA. The posterior hippocampus showed strong connectivity with bilateral regions of HVIIA and the vermal portion of lobule V. Across ageing, similar reductions in connectivity were observed between the left, right, and anterior hippocampus and the border of cerebellar lobules HV and HVI. Minimal age-related connectivity changes were observed between the posterior hippocampus and the cerebellum.

## Results

To examine hippocampal-cerebellar functional connectivity, we analysed resting-state fMRI data from the CamCAN study[32]. The available dataset contained 653 adult participants (323 males, 330 females, 18–87 years old, mean = 54.3, SD = 18.6; for more information about participants see Participants section in the Methods or see ref. 40). Participants in

the dataset were cognitively healthy (based on mini-mental state examination; see ref. 41), and did not have any neurological or psychiatric conditions. Following data preprocessing and denoising, including motion quality control, 479 participants (242 males, 237 females, 18–87 years old, mean = 50.7, SD = 18.2) were carried forward to seed-based connectivity analyses using the CONN toolbox (see Methods). Hippocampal seeds were defined using the Harvard-Oxford subcortical atlas combined with the Jülich subiculum ROI, and subdivided into anterior and posterior parts at the uncal apex ($y = –21$). For statistical inference, group-level whole-brain maps were thresholded with family-wise error correction ($p < 0.05$), and results were localised within the SUIT cerebellar atlas (see Methods).

### The hippocampus is functionally connected to widespread areas of the cerebellar cortex

We first examined patterns of cerebellar connectivity with left and right hippocampal seed ROIs (see Methods). As predicted by anatomical studies, both the left and right hippocampus showed bilateral connections to widespread areas of the cerebellum, including the vermis and lateral hemispheres. Both hemispheres showed strongest connections with the border of lobule HIV and HV, HIX, HX, and lobule HVIIA (medial parts of Crus II and laterally within the horizontal fissure at the junction of Crus I and Crus II; Fig. 1a, b).

To examine hemispheric differences, cerebellar functional correlations with left and right hippocampus were contrasted directly. The left hippocampus, compared to the right hippocampus, showed preferential connections with a contralateral region of lobule HVIIA (Crus I and peak in Crus II; number of voxels = 3063; $p < 0.001$), as well as lobule HIX (number of voxels = 181, $p < 0.001$; Fig. 1c). Similarly, the right hippocampal seed also showed a strong preferential correlation with the contralateral (i.e., left) region of lobule HVIIA (peak in Crus II; number of voxels = 178, $p < 0.001$; Fig. 1d). Peak cluster statistics for these hemispheric contrasts are shown in Table 1.

### Long-axis subdivisions of the hippocampus show both overlapping and distinct patterns of cerebellar connectivity

Next, we examined hippocampal-cerebellar functional connectivity along the longitudinal axis of the hippocampus. The anterior and posterior hippocampus showed connectivity to similar regions as those connected to the left and right hippocampus, including the border of lobule HIV and HV, vermal parts of lobule V, lobule HVIIA (bilaterally at the fissure separating Crus I and II), lobule HIX and HX (Fig. 2a, b).

However, direct contrasts revealed that the anterior hippocampus, compared to the posterior hippocampus, showed significantly greater functional correlations with lobule HI-HIV (number of voxels = 185, $p < 0.001$) and bilateral regions of lobule HVIIA, including right Crus II (number of voxels = 2778, $p < 0.001$; Fig. 2c). Note, this same region of Crus II also displays significantly greater connectivity with left versus right hippocampus (Fig. 1c). Additional clusters were also seen in extreme areas of right lobule HIX and HX, and bilateral HVIIB and HVIII. The posterior versus anterior hippocampus contrast showed significantly greater connectivity with widespread regions of the anterior cerebellum. Specifically, we observed a peak in the bilateral extremes of the Crus I region of lobule HVIIA (number of voxels = 4761; $p < 0.001$), which extended medially into vermal areas of lobule V (see Fig. 2d). Further small clusters were found in the lateral extremes of lobule HVIIIA, bilaterally, which continued along the border of HVIIIB and HIX. Peak cluster statistics for these direct long-axis contrasts are shown in Table 1.

### Hippocampal-cerebellar functional connectivity is reduced in ageing

Finally, we examined the effect of age on hippocampal-cerebellar functional connectivity (see Methods). We observed that several cerebellar regions showed negative correlations between age and hippocampal-cerebellar functional connectivity, that is, reduced connectivity in older participants.

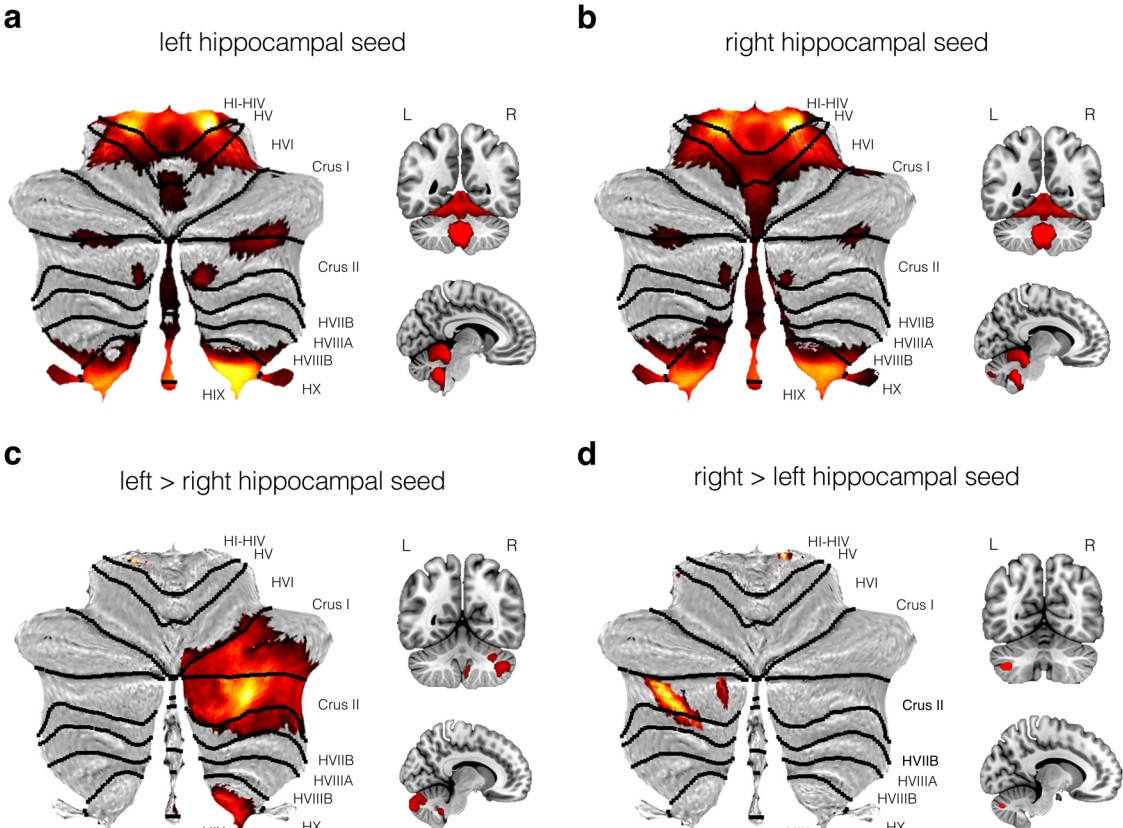

**Fig. 1 | Cerebellar regions showing significant functional connectivity with left and right hippocampus.** Thresholded SPM {T} maps ($p < 0.05$) are overlaid on cerebellar flatmaps using SUIT (Diedrichsen et al.[103]), as well as on coronal and mid-sagittal sections of the standard MNI T1 2 mm template. Parametric maps show the main effects of **a** left and **b** right hippocampal seeds, and the contrasts of **c** left > right and **d** right > left hippocampal seed connectivity with the cerebellum. Statistical parametric maps are depicted in a red-yellow colour scale with brighter colours indicating increased functional connectivity.

For the left hippocampal seed, we saw large age-related reductions in the depths of the right primary fissure that separates cerebellar lobules HV and HVI (number of voxels = 382; $p < 0.001$), which extended into adjacent parts of lobule HVIIA (Crus I; see Fig. 3a). A similar spatial pattern of connectivity alterations was seen for the right hippocampus but also revealed a strong cluster in the contralateral region of lobule HVI (number of voxels = 71; $p < 0.001$; see Fig. 3b). No differences were observed when directly contrasting left and right hippocampus.

Next, we explored whether these age-related effects within the cerebellum differed across different hippocampal long-axis subdivisions. Notably, we found that the anterior hippocampus showed widespread age-related connectivity reductions within the cerebellum, and these were present primarily within the primary fissure between lobules HV and HVI (number of voxels = 837; $p < 0.001$; see Fig. 3c). In contrast, the posterior hippocampus showed minimal age-related changes in functional connectivity, with very small suprathreshold clusters observed in dispersed areas of the cerebellum, including lobules HV and HVI, with a peak in lobule HIX (number of voxels = 1; $p = 0.026$; see Fig. 3d). However, no age-dependent connectivity differences were observed when contrasting anterior and posterior hippocampus. Cluster statistics for the ageing analyses are shown in Table 2. Note, this general pattern of functional connectivity held when controlling for sex and cardiovascular markers (i.e., BMI; see Supplementary Results and Supplementary Fig. 1).

## Discussion

While prior evidence in nonhuman animals suggest that a close functional interaction exists between the hippocampus and cerebellum, there is a limited understanding of the nature and topography of this functional connection in humans, and how it might vary with age. In the current study,

we sought to close this gap by applying seed-based functional connectivity analyses to resting-state fMRI data collected in a large-scale lifespan cohort ($N = 479$ from CamCAN). Our results yielded several key findings. Firstly, the shared bilateral functional connectivity between the left and right hippocampus with cerebellum involved four areas of the cerebellar cortex bilaterally: (i) almost the whole of the anterior lobe, and in addition, the anterior bank of lobule HVI; (ii) lobule HVIIA, both the anterior and posterior bank in the mid-point of the horizontal fissure; (iii) a separate region in lobule HVIIA in Crus II, adjacent to a medial part of the anso-paramedian fissure; and (iv) lobules HIX and HX. We also considered laterality-related connectivity differences. When directly contrasting the left and right hippocampus, we found that the left hippocampus showed greater connectivity with contralateral lobules HVIIA (Crus I and Crus II) and HIX, whilst the right hippocampus showed greater connectivity with the contralateral lobule HVIIA (Crus II).

Second, we found that the anterior and posterior hippocampus were functionally connected with similar areas of the cerebellum as those associated with the left and right hippocampus. However, when directly contrasting functional connectivity between anterior and posterior hippocampus, we found that each was preferentially correlated with distinct parts of the cerebellar cortex. The anterior compared to posterior hippocampus contrast revealed largely medial cerebellar connectivity dominated by areas posterior to the horizontal fissure, including the mid-portion of lobule HVIIA, extending into more medial parts of HVIIB and HVIII, and small areas of lobule HIX and HX. In contrast, the posterior compared to anterior hippocampus contrast revealed connectivity with areas of the cerebellar cortex that were largely anterior to the horizontal fissure, including the vermal portion of lobules I-VII, and hemispherical portions of lobules HI to HVIIA, including Crus I.

**Table 1 | Significant clusters identified within the cerebellum from all hippocampal seeds**

| Contrasts | Cerebellar lobule | Peak coordinate (mm) | Probabilistic value | z | t |
|---|---|---|---|---|---|
| Left > right hippocampus | Right lobule HVIIA (Crus II) | 36 -80 -42 | Right Crus II (97%) | Inf | 10.7 |
| | | 28 -84 -32 | Right Crus I (80%) | Inf | 10.14 |
| | | | Right Crus II (19%) | | |
| | | 28 -76 -44 | Right Crus II (76%) | inf | 9.23 |
| | Right lobule HIX | 4 -52 -48 | Right HIX (93%) | Inf | 8.28 |
| | | | Brain stem (1.2%) | | |
| | | 6 -48 -40 | Right HIX (83.2%) | 7.69 | 7.94 |
| | | | Brain stem (16.8%) | | |
| Right > left hippocampus | Left lobule HVIIA (Crus II) | -38 -64 - 44 | Left Crus II (87%) | 6.34 | 6.48 |
| | | | Left Crus I (3%) | | |
| | | -36 -70 -50 | Left Crus II (63%) | 5.96 | 6.07 |
| | | | Left VIIB (34%) | | |
| | | -28 -74 -50 | Left VIIB (63%) | 5.66 | 5.76 |
| | | | Left Crus II (36%) | | |
| | Right lobule HI-HIV | 12 -34 -20 | Right HI-HIV (60%) | 6.12 | 6.24 |
| | | | Brain stem (24.4%) | | |
| | Left lobule HVIIA (Crus II) | -12 -78 -34 | Left Crus II (87%) | 5.64 | 5.74 |
| | Right lobule HVIIIA | 32 -44 -56 | Right HVIIIA (28%) | 5.49 | 5.58 |
| | | | Right HVIIIB (6%) | | |
| | Left lobule VI | -36 -36 -38 | Left VI (62%) | 5.39 | 5.48 |
| | | | Left V (4%) | | |
| | Left lobule VIIB | -20 -76 -52 | Left VIIB (88%) | 5.21 | 5.28 |
| | | | Left Crus II (10%) | | |
| | Left lobule VIIB | -40 -54 -58 | Left VIIB (40%) | 5.19 | 5.27 |
| | | | Left VIIIA (26%) | | |
| | Right lobule HVIIIB | 20 -44 -56 | Right HVIIIB (89%) | 5.09 | 5.16 |
| | | | Right HIX (2%) | | |
| | Left lobule HVIIB | -32 -70 -58 | Left HVIIB (77%) | 5.03 | 5.1 |
| | | | Left HVIIIA (6%) | | |
| Anterior > posterior hippocampus | Right lobule HI-HIV | 12 -42 -28 | Right HI-HIV (47%) | Inf | 13.07 |
| | | | Brain stem (4%) | | |
| | | 20 -36 -30 | Right HI-HIV (37%) | Inf | 12.11 |
| | | | Brain stem (15.1%) | | |
| | | 22 -46 -30 | Right HV (17%) | Inf | 10.28 |
| | | | Right VI (14%) | | |
| | Right lobule HVIIA (Crus II) | 28 -84 -44 | Right Crus II (92%) | Inf | 12.13 |
| | | 36 -80 -40 | Right Crus II (93%) | Inf | 11.99 |
| | | | Right Crus I (6%) | | |
| | | 14 -84 -46 | Right Crus II (79%) | Inf | 10.26 |
| | | | Right HVIIB (7%) | | |
| | Brain stem | -8 -42 -28 | Brain stem (69.7%) | Inf | 11.48 |
| | | | Left HI-HIV (19.0%) | | |
| | | -18 -34 -30 | Brain stem (48.6%) | Inf | 11.23 |
| | | | Left HI-HIV (20%) | | |
| | | -26 -42 -34 | Left HVI (7%) | 7.68 | 7.93 |
| | | | Left HV (5%) | | |
| | Right lobule HIX | 6 -46 -36 | Right HIX (32%) | Inf | 8.96 |
| | | | Brain stem (11.3%) | | |
| | | -2 -46 -34 | Vermal X (39%) | 7.37 | 7.59 |
| | | | Brain stem (2.1%) | | |

**Table 1 (continued) | Significant clusters identified within the cerebellum from all hippocampal seeds**

| Contrasts | Cerebellar lobule | Peak coordinate (mm) | Probabilistic value | z | t |
|---|---|---|---|---|---|
| | | 0 -48 -44 | Right HIX (42%) | 6.43 | 6.57 |
| | | | Left IX (14%) | | |
| | Right lobule VIIIB | 12 -40 -60 | Right HVIIIB (12%) | 5.19 | 5.26 |
| | Right lobule HVIIA (Crus I) | 46 -58 -34 | Right Crus I (93%) | 5.05 | 5.12 |
| Posterior > anterior hippocampus | Left lobule HVIIA (Crus I) | -44 -42 -34 | Left Crus I (49%) | Inf | 16.6 |
| | | | Left HVI (15%) | | |
| | | 46 -40 -34 | Right Crus I (22%) | Inf | 15.96 |
| | | | Right HVI (3%) | | |
| | | 0 -60 -6 | Left HV (47%) | Inf | 15.8 |
| | | | Right HV (42%) | | |
| | Left lobule HVIIB | -30 -68 -48 | Left HVIIB (70%) | 7.78 | 8.04 |
| | | | Left Crus II (17%) | | |
| | | -40 -60 -42 | Left Crus II (49%) | 6.92 | 7.1 |
| | | | Left Crus I (48%) | | |
| | Left lobule HIX | -10 -52 -50 | Left HIX (71%) | 6.55 | 6.7 |
| | | | Left HVIIIB (2%) | | |
| | Right lobule HVIIA (Crus I) | 50 -62 -26 | Right Crus I (64%) | 6.33 | 6.47 |
| | | | Temporal Occipital Fusiform (3%) | | |
| | Right lobule HVIIA (Crus I) | 20 -90 -26 | Right Crus I (54%) | 5.19 | 5.27 |
| | | | Right Crus II (23%) | | |

Peak coordinates (MNI 2 mm space), z- and t-values are reported for all clusters. All statistics are FWE-corrected at *p* < 0.05. Negative x-coordinates indicate the left hemisphere. Inf indicates large values that surpass the statistical threshold. Probabilistic values were derived using the SPM Anatomy Toolbox[109].

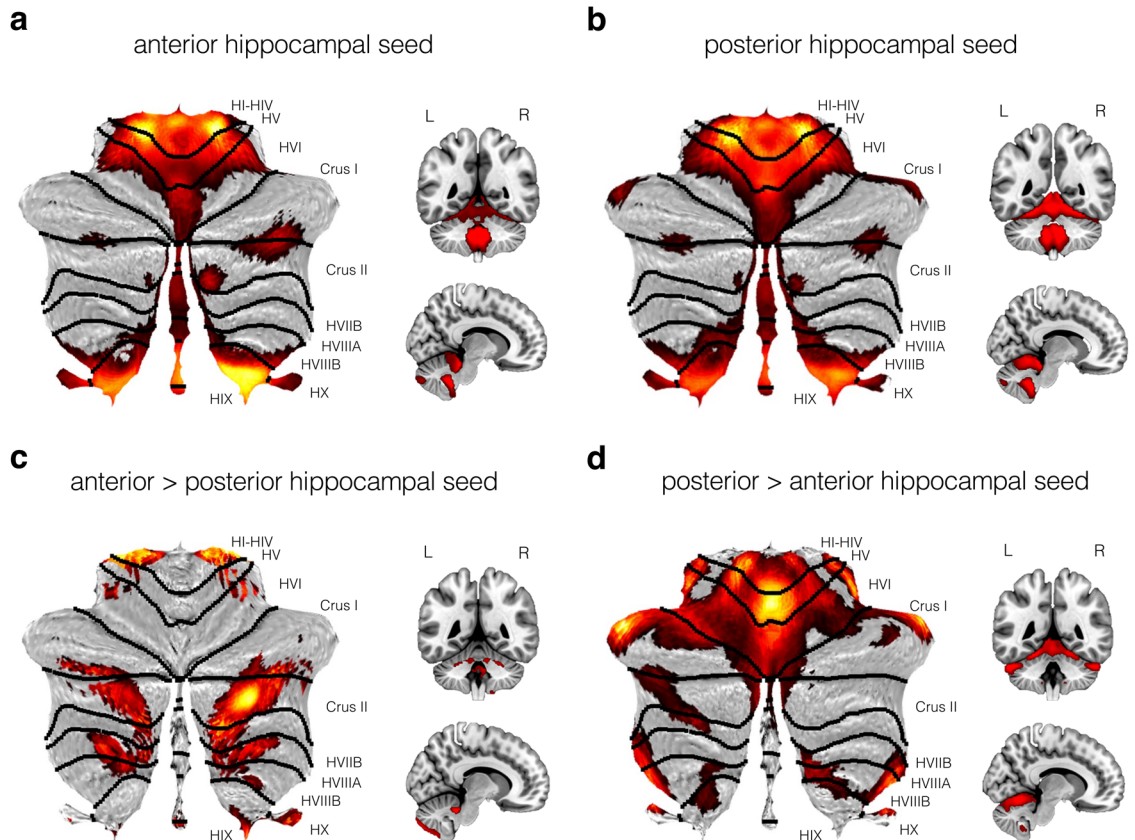

**Fig. 2 | Cerebellar regions showing significant functional connectivity with anterior and posterior hippocampal seeds.** Thresholded SPM {T} maps (*p* < 0.05) are overlaid on cerebellar flatmaps, as well as coronal and mid-sagittal slices. Parametric maps display the main effects of **a** anterior and **b** posterior hippocampal seeds, and the contrasts of **c** anterior > posterior and **d** posterior > anterior hippocampal seed connectivity with the cerebellum. Statistical parametric maps are depicted in a red-yellow colour scale with brighter colours indicating increased functional connectivity.

**Fig. 3 | Age-related decreases in functional connectivity between key hippocampal seeds and cerebellar cortex.** Thresholded SPM {T} maps ($p < 0.05$) are overlaid on cerebellar flatmaps, and show the main effect of age on connectivity between the cerebellum and **a** left, **b** right, **c** anterior, and **d** posterior hippocampal seeds. Statistical parametric maps are depicted in a blue-purple colour scale with darker colours indicating reduced functional connectivity with seed ROI with age.

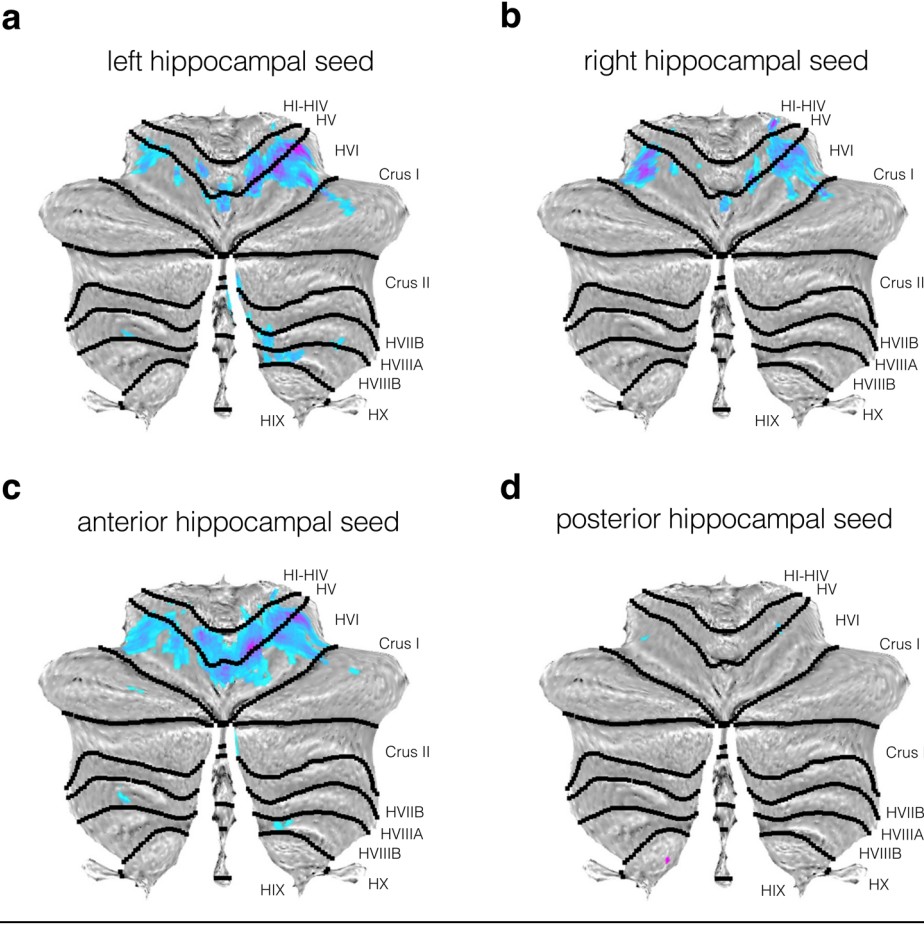

Finally, we observed age-related decreases in hippocampal-cerebellar connectivity that were confined largely to areas of the anterior lobe. Similar patterns were observed for left and right hippocampus, whereby decreases were observed in lobule HVI, and around the right primary fissure. Anterior hippocampus showed age-related decreases in connectivity with those same areas of the cerebellar cortex, whereas the posterior hippocampus revealed no appreciable effects in any part of it.

Previous electrophysiological studies have suggested a strong functional interaction exists between the hippocampus and cerebellum, based on the effects of cerebellar disruption on hippocampal place cell firing[16,17] and hippocampal-cerebellar synchronous oscillatory activity[22]. The current findings support that these functional interactions exist in the human brain but, importantly, elucidates the topography/spatial distribution of these functional interactions in the cerebellum. The finding that the hippocampus is significantly functionally connected with cerebellar lobules HIV, HV, HVI, and bilateral regions of lobule HVIIA (Crus I and Crus II), aligns with prior electrophysiological findings[22,42,43] and anatomical work in mice[22,44]. For instance, hippocampal field potentials have been shown to be modulated, most strongly, by stimulation of (vermal) lobules IV–V[43], and phase-locked theta coherence has been observed between dorsal hippocampus and cerebellar lobule VI and VIIA (Crus I)[22]. Likewise, the combination of c-Fos imaging and graph theory suggest that two cerebellar networks—one including lobule IV–V, VI, lobule VIIA (Crus I) and other including lobule IX, X—are differentially connected with dorsal hippocampal CA1 neurons during spatial exploration[42]. Notably, these same cerebellar lobules are also functionally correlated with left, right, anterior and posterior hippocampal seeds in this study (see Figs. 1a, b, 2a, b).

Our findings also align with previous tracing studies. For example, tracing injections into the dentate gyrus of the mouse hippocampus identified polysynaptic connectivity with lobule HIV and lobule HV[44]. Further,

retrograde tracing studies also identified inputs to the left hippocampus from lobule HVI and lobule HVIIA (Crus I)[22]— all regions which were also found be functionally connected with the hippocampus in the current study, some of these even showing preferential connections with specific hippocampal regions (e.g., lobule HIV, HV, and HVI with posterior hippocampus).

Our findings also provide insights into hippocampal-cerebellar connectivity in humans. Notably, in contrast to prior work in nonhuman species (discussed above), the strongest connectivity in lobule HVIIA was found in Crus II rather than the more commonly reported Crus I[22]. This was particularly evident when contrasting anterior and posterior hippocampus—a comparison which has not been made in the animal studies. While each long-axis subdivision showed broadly similar patterns of cerebellar connectivity when considered independently (and indeed showed patterns akin to the hippocampus as a whole), direct contrasts revealed different patterns of functional correlation within the cerebellum. Specifically, we found that the anterior hippocampus was more strongly correlated with Crus II (extending into lobule HVIIB and HVIII). In comparison, the posterior hippocampus showed strong functional correlations with lobule HVIIA and HV (extending into lobule VI).

Previous fMRI studies have implicated lobule HVIIA (Crus II) in nonmotor functions such as first- and second-order rule learning[45,46], creative thinking[47], social/emotional processing[48,49] and autobiographical memory[50]. This is also consistent with its closer functional relationship with a "core" posteromedial cortical network (often termed the 'default mode network')[51], which—alongside the anterior hippocampus[52] — is thought to support a range of constructive cognitive processes, including scene construction, autobiographical memory, and theory of mind[53,54]. In contrast, a region more strongly associated with posterior hippocampus (lobule V) is integral to the motor control system. It is interconnected with cortical motor areas[11],

**Table 2 | Significant cerebellar clusters displaying significant age-related reductions with hippocampal seeds**

| Seed ROI | Peak cerebellar lobule | Peak coordinate (mm) | Probabilistic value | z | t |
|---|---|---|---|---|---|
| Left hippocampus | Right lobule HVI | 32 -42 -26 | Right HVI (61%) | 6.7 | 6.87 |
| | | | Right HV (11%) | | |
| | | 14 -58 -16 | Right HV (55%) | 6.52 | 6.67 |
| | | | Right HVI (45%) | | |
| | | 22 -50 -22 | Right HVI (68%) | 6.45 | 6.59 |
| | | | Right HV (31%) | | |
| | Vermal lobule VI | 0 -66 -16 | Vermis VI (32%) | 5.89 | 6 |
| | | | Right HV (26%) | | |
| | Left lobule HV | -10 56 -14 | Left HV (95%) | 5.85 | 5.96 |
| | | | Left HI-HIV (2%) | | |
| | Vermal lobule VIIIA | 6 -64 -30 | Vermis VIIIA (23%) | 5.58 | 5.67 |
| | | | Right HVIIIA (8%) | | |
| | Left lobule HVI | -10 -64 -14 | Left HVI (83%) | 5.52 | 5.62 |
| | | | Left HV (16%) | | |
| | Right lobule HVIIIA | 14 -66 -44 | Right HVIIIA (56%) | 5.48 | 5.57 |
| | | | Right HVIIB (15%) | | |
| | | 16 -58 -46 | Right HVIIIB (46%) | 5.45 | 5.54 |
| | | | Right HIX (13%) | | |
| | Left lobule HVI | -30 -48 -26 | Left HVI (94%) | 5.46 | 5.55 |
| | | | Left HV (5%) | | |
| | | -32 -38 -28 | Left HVI (46%) | 5.27 | 5.35 |
| | | | Left HV (23%) | | |
| | Left lobule HVI | -22 -56 -20 | Left HVI (92%) | 5.44 | 5.52 |
| | | | Left HV (7%) | | |
| | Left lobule HVIIIA | -24 -56 -48 | Left HVIIIA (47%) | 5.17 | 5.25 |
| | | | Left HVIIIB (39%) | | |
| | Right lobule HVIIIA | 28 -50 -48 | Right HVIIIA (62%) | 5.11 | 5.18 |
| | | | Right HVIIIB (17%) | | |
| | Right lobule HVIIB | 6 -72 -42 | Right HVIIB (51%) | 5.01 | 5.08 |
| | | | Right HVIIIA (19%) | | |
| | Right lobule HI-HIV | 22 -32 -24 | Right HI-HIV (30%) | 4.98 | 5.04 |
| | | | Parahippocampal gyrus (23%) | | |
| Right hippocampus | Left lobule HVI | -32 -52 -24 | Left HVI (95%) | 6.21 | 6.34 |
| | Right lobule HV | 12 -60 -16 | Right HV (56%) | 6.12 | 6.24 |
| | | | Right HVI (42%) | | |
| | | 14 -52 -18 | Right HV (88%) | 5.19 | 5.27 |
| | | | Right HVI (5%) | | |
| | Right lobule HVI | 32 -42 -26 | Right HVI (61%) | 6.07 | 6.19 |
| | | | Right HV (11%) | | |
| | | 24 -48 -22 | Right HVI (68%) | 6.01 | 6.13 |
| | | | Right HV (30%) | | |
| | | 22 -30 -26 | Right HI-HIV (44%) | 5.92 | 6.04 |
| | | | Parahippocampal gyrus (16%) | | |
| | Vermal lobule VI | 0 -66 -16 | Vermis VI (32%) | 5.59 | 5.69 |
| | | | Right HV (26%) | | |
| | Left lobule HVI | -24 -54 -20 | Left HVI (90.1%) | 5.31 | 5.39 |
| | | | Left HV (6.9%) | | |
| | Vermal lobule VIIIA | 0 -74 -40 | Vermis VIIIA (56%) | 5.08 | 5.15 |
| | | | Vermis VIIB (8%) | | |
| | Left lobule HV | -10 -54 -14 | Left HV (89%) | 4.99 | 5.05 |
| | | | Left HI-HIV (8%) | | |

**Table 2 (continued) | Significant cerebellar clusters displaying significant age-related reductions with hippocampal seeds**

| Seed ROI | Peak cerebellar lobule | Peak coordinate (mm) | Probabilistic value | z | t |
|---|---|---|---|---|---|
| Anterior hippocampus | Right lobule HV | 12 -60 -16 | Right HV (56%) | 7.65 | 7.89 |
| | | | Right HVI (42%) | | |
| | | 32 -42 -26 | Right HVI (61%) | 7.41 | 7.63 |
| | | | Right HV (11%) | | |
| | | 0 -66 -16 | Vermis VI (32%) | 7.02 | 7.21 |
| | | | Right HV (26%) | | |
| | Left lobule HVI | -22 -56 -20 | Left HVI (92%) | 6.06 | 6.19 |
| | | | Left HV (7%) | | |
| | | -32 -50 -26 | Left HVI (98%) | 6.01 | 6.13 |
| | | | Left HV (2%) | | |
| | | -24 -46 -22 | Left HVI (49%) | 5.89 | 6 |
| | | | Left HV (48%) | | |
| | Left lobule HVIIIA | -24 -56 -48 | Left HVIIIA (47%) | 5.41 | 5.49 |
| | | | Left HVIIIB (39%) | | |
| | Left lobule HV | -18 -42 -18 | Left HV (71%) | 5.12 | 5.19 |
| | | | Left HI-HIV (16%) | | |
| | Right lobule HVIIIB | 14 -60 -44 | Right HVIIIB (33%) | 5.09 | 5.16 |
| | | | Right HIX (11%) | | |
| | Right lobule HVIIA (Crus I) | 48 -62 -26 | Right Crus I (83%) | 5.07 | 5.14 |
| | | | Temporal Occipital Fusiform (3%) | | |
| | Left lobule HV | -22 -36 -22 | Left HV (40%) | 5.07 | 5.14 |
| | | | Parahippocampal gyrus (24%) | | |
| | Left lobule HVIIA (Crus I) | -40 -72 -24 | Left Crus I (96%) | 5.03 | 5.1 |
| | Vermal lobule VIIIA | 6 -64 -30 | Vermis VIIIA (23%) | 5.02 | 5.09 |
| | | | Right HVIIIA (8%) | | |
| Posterior hippocampus | Left lobule HIX | -2 -54 -56 | Left HIX (53%) | 5.1 | 5.18 |
| | | | Right HIX (17%) | | |
| | Right lobule HVI | 24 -46 -22 | Right HVI (53%) | 4.99 | 5.06 |
| | | | Right HV (45%) | | |
| | Left lobule HVI | -32 -50 -26 | Left HVI (98%) | 4.98 | 5.04 |

Peak coordinates (MNI 2 mm space), z- and t-values are shown for all clusters. All statistics are FWE-corrected at *p* < 0.05. Negative x-coordinates indicate left hemisphere peak voxels. Probabilistic values were derived from the SPM Anatomy Toolbox[109].

receives limb proprioceptive afference via the spinocerebellar system[55], and contains fine-grained digit representations[56,57].

This differential connectivity of anterior and posterior hippocampus to functionally distinct regions of the cerebellum suggests that functional subdivisions within the hippocampal formation—which are considered to emerge partially from differences in neocortical connectivity[34,58,59] — are also mirrored in cerebellum. The cerebellum is thought to form both motor and non-motor loops with different cortical areas[39]. Our data suggest that the anterior hippocampus may interact preferentially with the cerebellar non-motor loop (consistent with its increased connectivity with prefrontal cortex)[60], whilst the posterior hippocampus may interact preferentially with the cerebellar motor loop. This distinction also aligns with current views of long-axis specialisation in the hippocampus, in which anterior hippocampus provides more abstract or coarse-grained representations in spatial and episodic memory, whereas the posterior hippocampus supports more fine-grained spatial processing[35,61].

Our findings also suggest that hippocampal-cerebellar connectivity decreases with age, and this was most evident in areas of cerebellar lobules HV, HVI, and their border. This finding dovetails with previous studies that have observed structural and functional changes in both the hippocampus and the cerebellum throughout ageing, making it plausible that this reflects, in part, age-related alterations in hippocampal-cerebellar interactions.

Indeed, the current findings show that the left, right, and anterior hippocampus show age-related connectivity reductions with areas of cerebellar lobules HV, HVI, and their border. This is in accordance with previous evidence that reported atrophy in lobule HVI during ageing[24,26]. Reduced grey matter volume in lobule HVI was also negatively associated with performance on a perspective-taking task[26] — a task which is thought to involve hippocampally-dependent allocentric representations[62]. Further, fMRI work in young adults reported increased but differential activation for place- versus sequence-based navigation in hemispheric lobule HVI[31]. The age-related reduction in hippocampal-lobule HVI connectivity seen here, therefore, may partly reflect age-related changes in the ability to adopt allocentric or place-based strategies during navigation, as observed in other studies[63,64].

In this context, it is important to note that while our results strongly support a close functional interaction between hippocampus and cerebellum, they cannot speak directly to how such connectivity supports spatial-mnemonic behaviour. In nonhuman species, lesions or genetic alterations to cerebellum has been shown to affect spatial learning and navigation, including on tasks highly sensitive to hippocampal lesions[14,65]. Further, mutant mouse models with postnatal degeneration of all cerebellar Purkinje cells — the key output of the cerebellum — showed impairments in the hidden (requiring the use of spatial strategies) but not the visually-cued

condition of the Morris Water Maze task[66]. Finally, unilateral removal of a cerebellar hemisphere impacts the spatial aspects in the Morris Water Maze task and inability to adapt behaviour to novel situations[67–70]. These pieces of evidence highlight that this connection might be important for spatial representations and the use of spatial strategies. This might be lateralised, expanding support for our findings that this organised topography might be involved in supporting specific aspects of behaviour. Further, there is growing evidence that the cerebellum directly influences hippocampal spatial representations, with disruption of cerebellar cells leading to alterations in hippocampal place cell firing[14].

Studies focused on the functional role of hippocampal-cerebellar interactions are limited in humans. However, one particular fMRI investigation found that distinct hippocampal-cerebellar connectivity patterns might relate to the application of different strategies during virtual navigation. For instance, it was found that the right hippocampus and contralateral lobule HVIIA (Crus I) co-activated during place-based navigation whilst left hippocampus and contralateral lobule HVIIA (Crus I) co-activated to support sequence-based navigation[31]. These same cerebellar regions were also strongly functionally correlated with the hippocampus in this study.

One possible account for these interactions is that the cerebellum supports the updating and re-organisation of hippocampally-based spatial representations (e.g., cognitive maps) via its use of forward models[15,71]. This describes a model, created from an internal command, that is an internal representation of the behaviour modified by sensory feedback[72]. Upon receiving sensory information that mismatches the internal representation, the cerebellum could support the re-computation of cognitive maps in the hippocampus when novel information is encountered.

An additional finding in the current study was that several hippocampal seeds showed strong functional correlations with regions of the 'vestibulocerebellum' — namely, lobules IX and X. While speculative, this functional interaction is consistent with prior work in rodents that has demonstrated the importance of vestibular information for the accuracy and stability of hippocampal representations[73], and aligns with theoretical models that suggest that the cerebellum supplies the hippocampus with information in an appropriate (world-centred) format to be used for allocentric navigation[14].

Functional connectivity approaches also cannot elucidate how the hippocampus and cerebellum are structurally connected. Whilst it was previously speculated that a direct route exists between these regions[20], recent work points towards a more indirect route via a number of relay stations. These include thalamic regions (e.g., laterodorsal and ventrolateral thalamus)[44], which receive input from the cerebellum[74] and project to areas that input to the hippocampus, such as the retrosplenial cortex[75]. Other relay stations include the septum and supramamillary nucleus, which are thought to be key regions involved in generating theta, and thus potentially relevant to theta coupling observed between hippocampus and cerebellum during navigation/exploration[22] and cerebellar-dependent eye-blink conditioning[76]. Other pathways include cerebellar outputs to the central-lateral thalamic nucleus which in turn project to the posterior parietal cortex (PPC) and retrosplenial cortex. These areas can project to the hippocampus via the entorhinal cortex[14]. The involvement of the PPC and retrosplenial cortex is plausible due to their role in spatial navigation. For example, the PPC in involved in integrating self-motion and contains cells that encode movement egocentrically[77,78]. Indeed, PPC has also shown functional connectivity with cerebellar lobules VIIA, Crus I and II[13]. The retrosplenial cortex is thought to be involved in the process of converting egocentric-to-allocentric and vice versa[79] — an important process for the use of different spatial navigation strategies. Further, it is thought to contain head-direction cells that rely on vestibular information[80] — posing it as an important mediating structure working alongside the vestibulocerebellum.

It may be possible to map structural pathways between cerebellar regions/nuclei and the human hippocampus by leveraging high-resolution diffusion MRI tractography. However, the fine-grained folial complexity of the cerebellum, coupled with the range and complexity of fiber populations

(e.g., through the midbrain), would require advanced MRI acquisition and modelling approaches[81], and potential validation data in other species. One study, for example, used probabilistic constrained spherical deconvolution tractography (which enables complex/crossing fibre populations to be better resolved) and identified tractography streamlines between the hippocampus and cerebellar lobules HVIII, IX, X, Crus I, Crus II, and the fastigial nucleus[82], though it will be critical to validate these findings using ultra-high field/gradient diffusion MRI.

Here, through the application of seed-based functional connectivity analyses within a large lifespan imaging cohort, we demonstrated that the human hippocampus is strongly functionally correlated with widespread areas of the cerebellar cortex, including hemispherical and vermal regions of lobules I–IV, V, VI, Crus I, Crus II, IX and X. This finding provides additional important support for the idea that the two key brain structures, which are classically considered to underpin distinct memory systems, may collaborate closely in the human brain. Further, we show that hippocampal-cerebellar connectivity decreases significantly across the lifespan, particular in cerebellar regions that are vulnerable to age-related structural atrophy (lobules HV and HVI). It will be important for future studies to examine this interaction during behaviour, which will inform the development of more detailed neurobiological models of (spatial) learning and memory that incorporate these distinct regions, as well as examine the vulnerability of this functional link in neurodegenerative disorders (e.g., Alzheimer's disease).

## Methods

### Participants

We used structural and functional MRI data from the Cambridge Centre of Ageing and Neuroscience (CamCAN) study[32]. This dataset contained 653 participants (323 males, 330 females, 18–87 years old, mean = 54.3, SD = 18.6). About fifty males and females were collected from each decile (deciles: 18–27 years, 28–37 years, 38–47 years, 48–57 years, 58–67 years, 68–77 years, 78–87 years; for more information about participants see ref. 40). Participants in the dataset were cognitively healthy, assessed via a score above 25 on the mini-mental state examination[41], and did not have any neurological or psychiatric conditions. Written informed consent for the database was obtained in accordance with the Cambridgeshire Research Ethics Committee, and these secondary data analyses were conducted in accordance with Royal Holloway, University of London Ethics Committee processes. All ethical regulations relevant to human research participants were followed.

### Neuroimaging data acquisition

All CamCAN MRI data was acquired using a 3 T Siemens TIM Trio Scanner at the Medical Research Council (UK) Cognition and Brain Science Unit using a 32-channel head coil. The data used in this study forms part of a larger scanning protocol (see https://camcan-archive.mrc-cbu.cam.ac.uk/dataaccess/ or[32] for more detail). High-resolution structural images were obtained using a T1-weighted magnetisation-prepared rapid sequence (MPRAGE; TE = 2.99 ms; TR = 2250 ms; TI = 900 ms; voxel size = 1 × 1 × 1 mm; field-of-view = 256 × 240 × 192 mm; flip angle = 9°). Resting-state fMRI data was acquired using a T2*-weighted gradient echo planar image (EPI) sequence. Rest (resting in the scanner with eyes closed) consisted of 261 volumes and lasted 8 min and 40 s (32 slices; TE = 30 ms, TR = 1970ms, voxel size: 3 × 3 × 4.44 mm; field-of-view: 192 × 192 mm).

### Data preprocessing and denoising

Initially, functional and structural MRI data from a random sample of approximately 25% of cases were visually inspected for image quality and motion artefacts by KA (with support from CJH). Functional MRI images were pre-processed and denoised using the standard pipeline in the CONN toolbox (RRID:SCR_009550; version 22.a)[83] and SPM12 (RRID: SCR_007037; version 12.7771)[84] running on Matlab 2022b (The Mathworks Inc, Natick, Massachusetts, USA). Functional MRI data were first realigned and unwarped using SPM12[85]. For this, all scans were co-registered to the first volume using a least-squares approach and a 6-parameter rigid body

transformation[86]. These were then resampled using b-spline interpolation to correct for motion and magnetic susceptibly interactions. Temporal misalignment between slices was corrected using the SPM12 slice-timing correction procedure[87,88], which involved sinc temporal interpolation to resample each BOLD timeseries slice to a common mid-acquisition time. Potential outlier scans (based on motion and global signal fluctuation) were identified using conservative (95th percentile) outlier parameters in ART (Artifact Detection Tools)[89]. Specifically, volumes with framewise displacement above 0.5 mm, or global BOLD signal changes above 3 SDs, were flagged as outliers[90,91]. A reference mean BOLD image was then computed for each subject by averaging all scans, excluding outliers. Following this, functional and structural images were separately normalised into standard MNI space and segmented into grey matter, white matter, and CSF 'tissue' types using the SPM12 unified segmentation and normalisation algorithm[92,93]. Functional and structural images were then resampled to 2 mm and 1 mm isotropic voxels, respectively, following a direct normalisation procedure[90,94] with the default IXI-549 tissue probability map template. Finally, functional data were smoothed using a Gaussian kernel of 5 mm full-width half-maximum (FWHM)[95].

Following preprocessing, the fMRI data were next denoised using the default pipeline in the CONN toolbox[96]. This involved regressing out noise components using an anatomical component-based noise correction procedure (aCompCor), which included noise components from white matter (5 noise components), CSF (5 noise components), motion parameters (3 translational and 3 rotational and their first order derivatives)[97], outliers volumes derived from scrubbing (38 factors)[91], effect of rest and its first order derivatives (2 factors; default setting which removes residual trends/instabilities only at the beginning of the timeseries). These were followed by a bandpass frequency filtering of the BOLD timeseries[98] between 0.01 Hz and 0.09 Hz (e.g., ref. [99]), which filters low frequencies (e.g., physiological noise). From the number of noise terms included in this denoising strategy, the effective degrees of freedom of the BOLD signal after denoising were estimated to range from 62.1 to 74.1 (average 70.5) across all subjects[90] (see ref. [100] for a full description of quality control measures, including degrees of freedom).

Following quality assurance, 479 participants (242 males, 237 females, 18–87 years old, mean = 50.7, SD = 18.2) were entered into the seed-based connectivity analysis. The remaining cases were rejected for reasons such as having more than 10% of invalid scans, motion above 0.5 mm, etc.

### Anatomical methods

Regions of interests (ROIs) for the hippocampus were defined using probabilistic atlases derived from anatomical segmentations. Hippocampal seed ROIs were created by combining the hippocampal ROI from the Harvard-Oxford subcortical atlas and the subiculum ROI from the Jülich histological atlas[101] ensuring that the hippocampal ROI was extended medially incorporating the subicular complex (see ref. [102]). The Harvard-Oxford atlas ROI was thresholded at 50% and the Jülich atlas ROI

thresholded at 75% to ensure both ROIs were constrained to grey matter and did not extend into adjacent regions. Using this method, left and right hippocampal ROIs were defined (Fig. 4a). For the long-axis analysis, the hippocampal ROIs were split into anterior and posterior zones arbitrarily at the uncal apex[1,35], corresponding to MNI slice $y = -21$ (see Fig. 4b for segmentation of the left hippocampus).

Functional correlations within the cerebellum were visualised and interrogated using the Spatially Unbiased Infratentorial Template (SUIT)[103]. As well as containing a detailed probabilistic atlas of the cerebellum, SUIT also contains a cerebellar flatmap, allowing for visualisation of functional connectivity maps within and across cerebellar lobules/subregions. The nomenclature of Larsell and Jansen[104] was used to characterise cerebellar cortical results.

### MRI analysis

Following denoising (see above), a first-level seed-based connectivity analysis was conducted using the CONN toolbox. Here, the BOLD timeseries for each hippocampal seed (left hippocampus, right hippocampus, anterior hippocampus, posterior hippocampus), as well as confound regressors (e.g., white matter and CSF), were entered as first-level covariates into general linear models (GLMs). Functional connectivity between the seeds and every other voxel in the brain was represented by the Fisher-transformed bivariate correlation coefficient ($r$-to-$Z$) from a weighted GLM. At the first level, we conducted eight analyses whereby four of them related to single seed and four of them to between-seed contrasts. Here we considered the issue of shared variance, and this was minimised by the use of separate GLMs for each seed which avoids multicollinearity between seeds in the same model. For the single seed analyses, the left, right, anterior, and posterior hippocampus design matrices contained a single column representing all subjects, allowing the investigation of seed-to-voxel connectivity for each ROI in isolation.

To examine hemispheric and long-axis differences in hippocampal connectivity with the cerebellum, we specified between-seed contrasts (referred to as between-source contrasts in CONN). Here, the design matrices included two columns representing each seed in each contrast. Contrasts compared the connectivity of left and right hippocampus (contrast vectors: 1 -1 and -1 1). Others compared the connectivity of anterior and posterior hippocampus (contrast vectors: 1 -1 and -1 1). For both single- and between-seed analyses, first-level beta or contrast images were carried to the second-level for one-sample $t$-tests. To explore age-related variation in hippocampal-cerebellar functional correlations, we also respecified these GLMs with age as an additional subject-level effect, firstly as a regressor-of-no-interest (contrast vector: 1 0), and then as a regressor-of-interest (contrast vector: 0 1) into a bivariate regression analysis. For these contrasts, age was mean-centred and to ensure that we captured anticorrelations, we applied a contrast vector of 0 -1 to the demeaned age data.

To additionally control for cardiovascular/metabolic health, which has been shown to influence functional connectivity[105–107], we respecified GLMs of age for a subset of the participants (due to unavailability of BMI for the full

---

**Fig. 4 | Hippocampal regions-of-interest. a** The left hippocampal ROI made from combining the hippocampus from the Harvard-Oxford atlas and the subicular complex from the Jüelich Histological atlas; **b** The left and right hippocampal ROI split into anterior and posterior hippocampus using the uncal apex for a landmark-based segmentation.

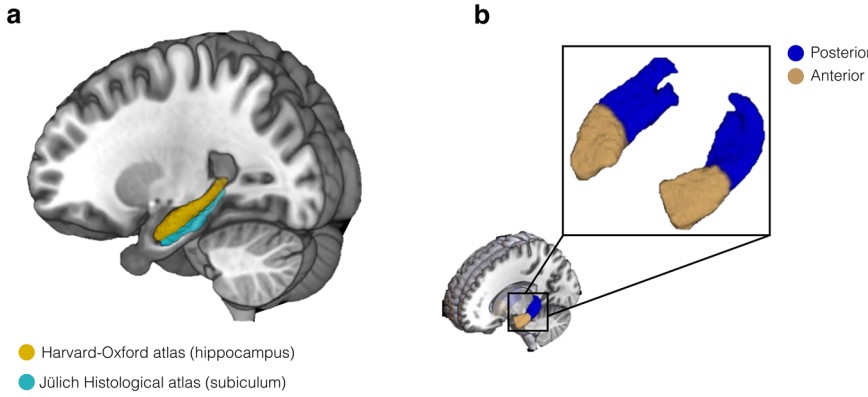

number of participants) with BMI (demeaned) and sex (binarized, 1 = female, 0 = male) as covariates-of-no-interest (contrast vector: 0 -1 0 0). We also conducted GLMs with BMI (contrast vectors: 0 0 -1 0 and 0 0 1 0) and sex (contrast vectors: 0 0 -1 0 and 0 0 1 0) as covariates-of-interest, controlling for age. These are reported in the Supplementary Results and visualised in Supplementary Fig. 1.

The unthresholded group-level $t$-statistic maps (as well as between-source contrast maps) from CONN were interrogated in SPM. These were thresholded using a family-wise error correction $p$-FWE < 0.05 based on Random Field Theory[108]. The thresholded images were then loaded into SUIT to localise and visualise significant clusters within cerebellar sub-regions. This produced cerebellar flatmaps showing suprathreshold connectivity strength within and across cerebellar lobules.

### Statistics and reproducibility
To statistically interpret functional connectivity between our hippocampal seeds (left, right, anterior and posterior) and the cerebellum, we conducted seed-based functional connectivity analyses using CONN toolbox (see MRI analysis). The hippocampus was defined using the Harvard-Oxford sub-cortical structural atlas, combined with the subiculum ROI from the Jülich histological atlas (to ensure more medial coverage). Anterior and posterior subdivisions were then extracted using the uncal apex ($y = -21$) as a split-ting point. To characterise the distribution of functional correlations with respect to cerebellar anatomy, we used the SUIT cerebellar flamap[103]. Fol-lowing the preprocessing and denoising of functional data (see 'Data pre-processing and denoising'), we calculated the correlation between each seed's timecourse and every other voxel in the brain. This was represented by the Fisher-transformed bivariate correlation coefficient ($r$-to-Z) from a weighted GLM. Second-level analyses were conducted in separate GLMs for each seed region where individual participant connectivity maps were entered into group-level models to characterise the connectivity of each seed to the whole brain, as well as the contrast between different seeds. These models were also respecified with age as a covariate of interest, as well as a covariate of no interest. For statistical inference, the unthresholded group-level whole-brain $t$-statistic maps and between-seed contrast maps were thresholded using a family-wise error correction $p$-FWE < 0.05 based on Random Field Theory[108]. The thresholded images were then loaded into SUIT to localise (and visualise) significant clusters within cerebellar subregions.

### Reporting summary
Further information on research design is available in the Nature Portfolio Reporting Summary linked to this article.

### Data availability
The CamCAN dataset is publicly available at https://camcan-archive.mrc-cbu.cam.ac.uk/dataaccess/.

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

## Acknowledgements

This work was supported a Sarah Parker Remond PhD studentship at Royal Holloway, University of London, and the Biotechnology and Biological Sciences Research Council (BBSRC) [BB/V010549/1]. For the purpose of open access, the authors have applied a Creative Commons Attribution (CC BY) licence to any Author Accepted Manuscript version arising. We would also like to thank Eleanor Alderman and Kieran Allen for their valuable advice during this project.

## Author contributions

Kavishini Apasamy: Conceptualisation, Formal Analysis, Data Curation, Visualisation, Writing—Original Draft, Writing—Reviewing and Editing. Samuel C Berry: Formal Analysis, Data Curation, Visualisation, Writing—Reviewing and Editing. Marie-Lucie Read: Formal Analysis, Data Curation, Visualisation, Writing—Reviewing and Editing. Narender Ramnani: Conceptualisation, Data Curation, Visualisation, Supervision, Writing—Reviewing and Editing. Carl J. Hodgetts: Conceptualisation, Visualisation, Supervision, Writing – Reviewing and Editing, Funding acquisition.

## Competing interests

The authors declare no competing interests.
