## [Transparent Peer Review file · Communications Biology]

Mapping hippocampal-cerebellar functional connectivity across the human adult lifespan

Corresponding Author: Dr Carl Hodgetts

This manuscript has been previously submitted at another journal. This document only contains information relating to versions considered at Communications Biology.

Version 0:

Reviewer comments:

Reviewer #1

(Remarks to the Author)

I congratulate the authors for well defined study of Mapping hippocampal-cerebellar functional connectivity across the adult lifespan.

The methodology included hippocampus as anterior and posterior regions rather than subdivisions or parcellations. I would have designed the study looking at functional connections between hippocampal subdivisions with cerebellar subdivisions and subsequent network analysis. Nonetheless the study has merits in current form.

While looking at age affect on functional connectivity, linear regression should age centre, binarize sex and add body-mass index as covariate. Age is affecting the connectivity but I would like to see motor component of cerebellum being investigated with negating BMI.

The table is comprehensive but should come as heatmap/matrix or go to supplementary section. It disrupts flow of reading.

Reviewer #2

(Remarks to the Author)

The paper "Mapping hippocampal-cerebellar functional connectivity across the adult lifespan" by Apasamy et al. explores the functional connectivity between the hippocampus and cerebellum in the human brain, particularly how it changes across the adult lifespan. Using resting-state fMRI data from the CamCAN dataset (N=479, aged 18–87), the authors identify significant hippocampal-cerebellar connectivity patterns, including hemispheric and longitudinal axis distinctions. They find that hippocampal connectivity is most pronounced in lobules HIV, HV, HVI, HVIIA (Crus I and II), HIX, and HX, with notable age-related declines in connectivity, particularly in lobules HVI and HVIIA. The study contributes novel insights into hippocampal-cerebellar interactions and their susceptibility to aging. The study presents compelling evidence of hippocampal-cerebellar connectivity and its age-related decline, with significant implications for understanding memory and spatial navigation processes in aging. However, improvements in interpretation, consideration of confounders, and additional discussion on functional implications are necessary before publication. Addressing these issues would enhance the impact and clarity of the manuscript.

1. The authors suggest hippocampal-cerebellar interactions may support spatial/mnemonic functions, but direct behavioral evidence is lacking. Including additional analyses (if behavioral data is available) and/or discussion on how these functional correlations may translate into cognitive or navigational impairments would strengthen the relevance.

2. Further, the discussion of anterior-posterior hippocampal connectivity could be expanded to integrate functional distinctions of the cerebellar regions identified. For example, are anterior hippocampal connections more associated with

cognitive functions beyond navigation, such as emotional regulation?

3. While motion correction and denoising strategies were implemented, residual age-related differences in head motion could contribute to connectivity reductions. A clearer discussion or additional control analyses addressing this issue would be beneficial. The authors should consider including head motion metrics (e.g., mean framewise displacement) as covariates in their statistical models to control for potential motion-related artifacts. Other relevant control variables that authors could include are cardiovascular health markers (e.g., blood pressure, BMI), cognitive performance scores, and white matter integrity metrics from diffusion MRI, which could better account for broader age-related changes that might influence functional connectivity.

4. Although the study provides strong functional connectivity data, it does not confirm the presence of direct anatomical pathways. Incorporating diffusion MRI evidence (if available) and/or discussing potential relay stations (e.g., thalamic or brainstem structures) more explicitly would improve the anatomical grounding of the findings.

5. The study uses a strict family-wise error correction, but some clusters are small (e.g., posterior hippocampal age effects). Reporting effect sizes or confidence intervals would help clarify the robustness of these findings.

6. The study primarily relies on the Harvard-Oxford and Jülich histological atlases for hippocampal delineation and the SUIT atlas for cerebellar mapping. While these are widely used, alternative atlases such as the Automated Anatomical Labeling (AAL) or the Schaefer 400 parcellation for functional connectivity may provide additional insights. For a finer-grained examination of hippocampal subfields, the FreeSurfer-based segmentation (e.g., the ex vivo hippocampal subfield atlas) could be utilized. Repeating analyses with one/some alternative atlases may refine the observed connectivity patterns and ensure robustness across different parcellation schemes.

Version 1:

Reviewer comments:

Reviewer #2

(Remarks to the Author)

The authors have addressed all my concerns. This manuscript is suitable for publication.

Response to reviewers

Reviewer 1

1.1 Comment: *I congratulate the authors for well defined study of mapping hippocampal-cerebellar functional connectivity across the adult lifespan.*

1.1 Response: We thank the reviewer for their positive appraisal and thoughtful suggestions (which are addressed below).

1.2 Comment: *The table is comprehensive but should come as heatmap/matrix or go to supplementary section. It disrupts flow of reading.*

1.2 Response: We thank the reviewer for this helpful suggestion to improve the readability of the paper. To directly address this, we have applied a red-blue colour map to the z-scores reported in Tables 1 and 2, which represents the “connectivity strength” between the hippocampus and cerebellar subregions. This ensure that essential detailed information in the tables while allowing for an intuitive visual representation of the connectivity strength. We are happy for any further changes if the editor or reviewers feel there is a more effective representation strategy.

1.3 Comment: *While looking at age affect on functional connectivity, linear regression should age centre, binarize sex and add body-mass index as covariate. Age is affecting the connectivity but I would like to see motor component of cerebellum being investigated with negating BMI.*

1.3 Response: We thank the reviewer for this thoughtful comment, particularly given the potential influence of cardiovascular and metabolic health markers (such as BMI) on functional connectivity (e.g., Rashid et al., 2023; Park et al., 2018; Beyer et al., 2017). In response, we have now included BMI (mean-centred) and sex (binarised) as covariates of no interest in our ageing analysis. Whilst controlling for BMI and sex led to modest changes in the intensity of certain connectivity hotspots, there were no appreciable differences between the original analysis and our re-analysis (see Figure 1). The anatomical patterns remained very similar across the two, so our conclusions do not change. These analyses are now included in the Supplementary Materials.

To further explore the relationship between these variables and hippocampal-cerebellar functional connectivity, we also tested positive and negative correlations with BMI and sex as covariates of interest. These analyses revealed negligible effects, with only a small number of voxels showing significance within cerebellar subregions (BMI: max $Z = 5.06$, 1 cluster, max cluster size = 1 voxel; Sex: max $Z = 5.56$, 3 clusters, max cluster size = 4 voxels).

For transparency and clarity, these additional results are reported in Supplementary Results, and changes have been made in the text of the Methods and Results section to reflect these. Finally, we presume that the reviewer meant “mean centre” instead of “age centre”, and can confirm that age was indeed mean-centred in all relevant analyses.

Figure 1. Regions in the cerebellar cortex showing age-related decreases in functional connectivity with hippocampal seed regions and when controlling for BMI and sex. Thresholded SPM $\{T\}$ maps overlaid on cerebellar flatmaps showing the effect of ageing between (a) left, (b) right, (c) anterior and (d) posterior hippocampal seeds and the cerebellum. The left of each panel shows the original flatmap (not controlled for BMI and sex) and the right shows the flatmap controlled for BMI and sex.

Reviewer 2

2.1 Comment: *The study contributes novel insights into hippocampal-cerebellar interactions and their susceptibility to aging. The study presents compelling evidence of hippocampal-cerebellar connectivity and its age-related decline, with significant implications for understanding memory and spatial navigation processes in aging.*

2.1 Response: We thank the reviewer for their positive comments on our manuscript and we hope that the incorporated changes outlined below enhance the clarity of the manuscript.

2.2 Comment: *The authors suggest hippocampal-cerebellar interactions may support spatial/mnemonic functions, but direct behavioral evidence is lacking. Including additional analyses (if behavioral data is available) and/or discussion on how these functional correlations may translate into cognitive or navigational impairments would strengthen the relevance.*

2.2 Response: We thank the reviewer for this important point. Unfortunately, behavioural data targeting spatial cognition is not available for the imaging dataset used in this study,

limiting our ability to relate hippocampal-cerebellar functional connectivity to spatial-mnemonic behaviour in the same participants. The resources required and volume of work involved in such an exercise means that this needs to be a separate piece of work. The reviewer makes a very important point, so as suggested, we have now expanded our discussion to include studies examining cognitive and navigational impairment following cerebellar lesions. The following has been added in lines 446-459:

“In nonhuman species, lesions or genetic alterations to cerebellum has been shown to affect spatial learning and navigation, including on tasks highly sensitive to hippocampal lesions (see Rochefort et al., 2013; Lefort et al., 2019). Further, mutant mouse models with postnatal degeneration of all cerebellar Purkinje cells – the key output of the cerebellum – showed impairments in the hidden (requiring the use of spatial strategies) but not the visually-cued condition of the Morris Water Maze task (Goodlett et al., 1992). Finally, unilateral removal of a cerebellar hemisphere impacts the spatial aspects in the Morris Water Maze task and inability to adapt behaviour to novel situations (Colombel et al., 2004; Petrosini et al., 1996; Leggio et al., 1999; Joyal et al., 1996). These pieces of evidence highlight that this connection might be important for spatial representations and the use of spatial strategies. This might be lateralised, expanding support for our findings that this organised topography might be involved in supporting specific aspects of behaviour. Further, there is growing evidence that the cerebellum directly influences hippocampal spatial representations, with disruption of cerebellar cells leading to alterations in hippocampal place cells firing. (e.g., Rochefort et al., 2013).”

2.3 Comment: *Further, the discussion of anterior-posterior hippocampal connectivity could be expanded to integrate functional distinctions of the cerebellar regions identified. For example, are anterior hippocampal connections more associated with cognitive functions beyond navigation, such as emotional regulation?*

2.3 Response: We thank the reviewer for this suggestion. We currently have a section relating these findings to functional distinctions within the cerebellum, but have extended this discussion as follows (shown in bold) (lines 402-428):

“Previous fMRI studies have implicated lobule HVIIA (Crus II) in non-motor functions such as first- and second-order rule learning (Balsters et al., 2013; Balsters and Ramnani, 2011), creative thinking (Gao et al., 2020), social/emotional processing (Guell and Schmahmann, 2020; Van Overwalle et al., 2020) and autobiographical memory (Addis et al., 2016). This is also consistent with its closer functional relationship with a “core” posteromedial cortical network (often termed the ‘default mode network’; Bucker et al., 2011), which – alongside the anterior hippocampus (Zeidman and Maguire, 2016) – is thought to support a range of constructive cognitive processes, including scene construction, autobiographical memory, and theory of mind (Spreng et al., 2009; Hassabis and Maguire, 2009). In contrast, a region more strongly associated with posterior hippocampus (lobule V) is integral to the motor control system. It is interconnected with cortical motor areas (Kelly and Strick, 2003), receives limb proprioceptive afference via the spinocerebellar system (see Oscarson, 1973), and contains fine-grained digit representations (Grodd et al., 2001; van der Zwaag et al., 2013).

This differential connectivity of anterior and posterior hippocampus to functionally distinct regions of the cerebellum suggests that functional subdivisions within the hippocampal formation – which are considered to emerge partially from differences in neocortical connectivity (Adnan et al., 2015; Aggleton, 2012; Dalton et al., 2019) – are also mirrored in cerebellum. The cerebellum is thought to form both motor and non-motor loops with different cortical areas (Ramnani et al., 2006). Our data suggest that the anterior hippocampus may interact preferentially with the cerebellar non-motor loop (consistent with its increased connectivity with prefrontal cortex; Cavada, 2000), whilst the posterior hippocampus may

interact preferentially with the cerebellar motor loop. This distinction also aligns with current views of long-axis specialisation in the hippocampus, in which anterior hippocampus provides more abstract or coarse-grained representations in spatial and episodic memory, whereas the posterior hippocampus supports more fine-grained spatial processing (Poppenk et al., 2013; Robin and Moscovitch, 2017)."

2.4 Comment: *Although the study provides strong functional connectivity data, it does not confirm the presence of direct anatomical pathways. Incorporating diffusion MRI evidence (if available) and/or discussing potential relay stations (e.g., thalamic or brainstem structures) more explicitly would improve the anatomical grounding of the findings.*

2.4 Response: We thank the reviewer for this important point and fully agree that considering anatomical connectivity would be important for interpreting our functional connectivity results. In line with the reviewer's suggestion, we already provide a detailed discussion of potential relay stations – and the challenges associated with diffusion MRI in this context – on p. 16.

Animal studies remain inconclusive regarding the existence of a direct anatomical pathway between the hippocampus and cerebellum. Furthermore, there is limited consensus – aside from studies such as Watson et al. (2019) – on which regions may act as relay stations between these structures. To ensure that diffusion MRI tractography outputs are both informative and anatomically plausible, it is essential to identify appropriate anatomical waypoints to constrain tracking and reduce false positives. Candidate relay structures include the pontine nuclei, which receive input from the cerebral cortex and project to the cerebellum. However, the fibre architecture within the midbrain is highly complex and requires advanced tractography algorithms (e.g., constrained spherical deconvolution) to resolve reliably.

The cerebellum itself presents several additional challenges compared to the major fibre bundles of the cerebral cortex (see Lundell & Steele, 2024, for a detailed discussion). First, its small and highly convoluted folial organisation necessitates much higher spatial and angular resolution to accurately capture fibre orientations and reduce partial volume effects, which require ultra-high field/gradient MRI data that is not available in the dataset used here. The highly complex and densely crossed fibre geometries in the pontine nuclei (the main relay station for layer 5 output neurons from the cerebral cortex) also mean that trajectories are difficult to resolve. These difficulties are further compounded by the cerebellum's anatomical position, which can exacerbate image distortions and eddy current-related signal loss.

We have now extended our discussion to capture these points (lines 501-509):

'It may be possible to map structural pathways between cerebellar regions/nuclei and the human hippocampus by leveraging high-resolution diffusion MRI tractography. However, the fine-grained folial complexity of the cerebellum, coupled with the range and complexity of fiber populations (e.g., through the midbrain), would require advanced MRI acquisition and modelling approaches (Lundell and Steele, 2024), and potential validation data in other species. One study, for example, used probabilistic constrained spherical deconvolution tractography (which enables complex/crossing fiber populations to be better resolved) and identified tractography streamlines between the hippocampus and cerebellar lobules HVIII, IX, X, Crus I, Crus II and the fastigial nucleus (Arrigo et al., 2014), though it will be critical to validate these findings using ultra-high field/gradient diffusion MRI.'

2.5 Comment: *While motion correction and denoising strategies were implemented, residual age-related differences in head motion could contribute to connectivity reductions. A clearer discussion or additional control analyses addressing this issue would be beneficial. The authors should consider including head motion metrics (e.g., mean framewise displacement) as covariates in their statistical models to control for potential motion-related artifacts. Other relevant control variables that authors could include are cardiovascular health markers (e.g., blood pressure, BMI), cognitive performance scores, and white matter integrity metrics from diffusion MRI, which could better account for broader age-related changes that might influence functional connectivity.*

2.5 Response: We thank the reviewer for raising this important point regarding motion-related artifacts. Our analysis applied the standard and widely recommended CONN (vR2020a) denoising pipeline. Specifically, it regresses six standard realignment parameters (three translational and three rotational) along with ART-based scrubbing regressors. Importantly, these are applied on a scan-to-scan basis, allowing for the removal of motion effect in the BOLD signal for each participant before connectivity is estimated – reducing its confounding influence on functional connectivity. This approach is well-validated and recommended (Whitfield-Gabrieli and Nieto-Castanon, 2012).

While we acknowledge the reviewers' suggestion to include mean framewise displacement as an additional second-level covariate, we note that the current preprocessing pipeline has been designed and implemented to minimise the effect of motion. Nonetheless, we appreciate that residual motion (following correction) may still influence our connectivity metrics/patterns, and we welcome further discussions on this.

In line with Reviewer 1's suggestion, we have also included BMI (mean-centred) and sex as covariates in our ageing models, and these are now reported in the Supplementary Materials. As noted in Response 1.3, the inclusion of these additional covariates had minimal impact on our results.

Regarding white matter integrity metrics, we refer the reviewer to Response 2.4, which outlines the significant methodological and conceptual challenges of tractography from cerebellar subregions.

As detailed in Response 2.2, behavioural data were not included due to the lack of relevant cognitive measures (i.e., spatial–mnemonic tasks) in this cohort. Furthermore, our study focuses specifically on the topography of hippocampal–cerebellar functional connectivity and its changes with age, rather than behavioural outcomes *per se*.

2.6 Comment: *The study uses a strict family-wise error correction, but some clusters are small (e.g., posterior hippocampal age effects). Reporting effect sizes or confidence intervals would help clarify the robustness of these findings.*

2.6 Response: We appreciate the reviewer's comment regarding the interpretation of small clusters, such as those observed in the posterior hippocampus in relation to age. We fully agree with the reviewer that effect sizes and confidence intervals can provide additional clarity regarding the robustness of results.

In this study, we applied strict family-wise error correction at the whole-brain level. As such, any cluster that survived this correction – regardless of the size – reflects a statistically robust effect.

Regarding effect sizes, the seed-based connectivity values from CONN are expressed as Fishers z-transformed correlation coefficients, which serve as effect size estimates. We believe that these provide a meaningful indication of the strength of the observed functional

correlations. Nonetheless, we appreciate the suggestion of the reviewer, and are happy to provide further clarification, if required.

2.6 Comment: *The study primarily relies on the Harvard-Oxford and Jülich histological atlases for hippocampal delineation and the SUIT atlas for cerebellar mapping. While these are widely used, alternative atlases such as the Automated Anatomical Labeling (AAL) or the Schaefer 400 parcellation for functional connectivity may provide additional insights. For a finer-grained examination of hippocampal subfields, the FreeSurfer-based segmentation (e.g., the ex vivo hippocampal subfield atlas) could be utilized. Repeating analyses with one/some alternative atlases may refine the observed connectivity patterns and ensure robustness across different parcellation schemes.*

2.6 Response: We thank the reviewer for these thoughtful suggestions. Our study was designed to conduct a focused investigation of hippocampal–cerebellar functional connectivity and its topographical organisation. For this reason, we used the SUIT atlas, which offers detailed and anatomically precise lobular cerebellar parcellation based on probabilistic estimates. While functional atlases such as the Schaefer 400 or AAL parcellation may be informative for broader cortical connectivity analyses, they do not provide additional insight into intra-cerebellar topography and therefore fall outside the scope of the present work.

With regard to hippocampal segmentation, we agree that finer-grained analyses using hippocampal subfields would be an important extension of this work. However, our resting-state fMRI data were acquired at a resolution of $3 \times 3 \times 4.44$ mm and further smoothed with a 5 mm full-width half-maximum kernel. This spatial resolution does not allow hippocampal subfields to be functionally delineated, which typically requires functional resolutions at or below 1.5 mm isotropic (see, e.g., Dalton et al., 2018; Hodgetts et al., 2017).

Moreover, although FreeSurfer-based hippocampal subfield segmentation from 1 mm T1-weighted images is widely used, such segmentations likely lack anatomical validity. In particular, T1-weighted images at this resolution do not allow visualisation of key internal landmarks, such as the stratum radiatum lacunosum moleculare (SRLM), which are essential for accurately distinguishing CA subfields from the dentate gyrus (see Wisse et al., 2020 for a detailed discussion of this issue).

In summary, our choice of atlases and parcellation schemes (e.g., anterior vs posterior hippocampus) reflects both the spatial resolution of our data and the anatomical precision required for our research question. We share the reviewer's interest in exploring hippocampal subfields in future work.

Response to reviewers

Reviewer 1

1.1 Comment: *I congratulate the authors for well defined study of mapping hippocampal-cerebellar functional connectivity across the adult lifespan.*

1.1 Response: We thank the reviewer for their positive appraisal and thoughtful suggestions (which are addressed below).

1.2 Comment: *The table is comprehensive but should come as heatmap/matrix or go to supplementary section. It disrupts flow of reading.*

1.2 Response: We thank the reviewer for this helpful suggestion to improve the readability of the paper. To directly address this, we have applied a red-blue colour map to the z-scores reported in Tables 1 and 2, which represents the “connectivity strength” between the hippocampus and cerebellar subregions. This ensure that essential detailed information in the tables while allowing for an intuitive visual representation of the connectivity strength. We are happy for any further changes if the editor or reviewers feel there is a more effective representation strategy.

1.3 Comment: *While looking at age affect on functional connectivity, linear regression should age centre, binarize sex and add body-mass index as covariate. Age is affecting the connectivity but I would like to see motor component of cerebellum being investigated with negating BMI.*

1.3 Response: We thank the reviewer for this thoughtful comment, particularly given the potential influence of cardiovascular and metabolic health markers (such as BMI) on functional connectivity (e.g., Rashid et al., 2023; Park et al., 2018; Beyer et al., 2017). In response, we have now included BMI (mean-centred) and sex (binarised) as covariates of no interest in our ageing analysis. Whilst controlling for BMI and sex led to modest changes in the intensity of certain connectivity hotspots, there were no appreciable differences between the original analysis and our re-analysis (see Supplementary Figure 1). The anatomical patterns remained very similar across the two, so our conclusions do not change. These analyses are now included in the Supplementary Information.

To further explore the relationship between these variables and hippocampal-cerebellar functional connectivity, we also tested positive and negative correlations with BMI and sex as covariates of interest. These analyses revealed negligible effects, with only a small number of voxels showing significance within cerebellar subregions (BMI: max $Z = 5.06$, 1 cluster, max cluster size = 1 voxel; Sex: max $Z = 5.56$, 3 clusters, max cluster size = 4 voxels).

For transparency and clarity, these additional results are reported in Supplementary Results, and changes have been made in the text of the Methods and Results section to reflect these. Finally, we presume that the reviewer meant “mean centre” instead of “age centre”, and can confirm that age was indeed mean-centred in all relevant analyses.

Supplementary Figure 1. Regions in the cerebellar cortex showing age-related decreases in functional connectivity with hippocampal seed regions and when controlling for BMI and sex. Thresholded SPM $\{T\}$ maps overlaid on cerebellar flatmaps showing the effect of ageing between (a) left, (b) right, (c) anterior and (d) posterior hippocampal seeds and the cerebellum. The left of each panel shows the original flatmap (not controlled for BMI and sex) and the right shows the flatmap controlled for BMI and sex.

Reviewer 2

2.1 Comment: *The study contributes novel insights into hippocampal-cerebellar interactions and their susceptibility to aging. The study presents compelling evidence of hippocampal-cerebellar connectivity and its age-related decline, with significant implications for understanding memory and spatial navigation processes in aging.*

2.1 Response: We thank the reviewer for their positive comments on our manuscript and we hope that the incorporated changes outlined below enhance the clarity of the manuscript.

2.2 Comment: *The authors suggest hippocampal-cerebellar interactions may support spatial/mnemonic functions, but direct behavioral evidence is lacking. Including additional analyses (if behavioral data is available) and/or discussion on how these functional correlations may translate into cognitive or navigational impairments would strengthen the relevance.*

2.2 Response: We thank the reviewer for this important point. Unfortunately, behavioural data targeting spatial cognition is not available for the imaging dataset used in this study, limiting our ability to relate hippocampal-cerebellar functional connectivity to spatial-mnemonic behaviour in the same participants. The resources required and volume of work involved in such an exercise means that this needs to be a separate piece of work. The reviewer makes a very important point, so as suggested, we have now expanded our discussion to include studies examining cognitive and navigational impairment following cerebellar lesions. The following has been added in lines 266-277:

“In nonhuman species, lesions or genetic alterations to cerebellum has been shown to affect spatial learning and navigation, including on tasks highly sensitive to hippocampal lesions (see Rochefort et al., 2013; Lefort et al., 2019). Further, mutant mouse models with postnatal degeneration of all cerebellar Purkinje cells – the key output of the cerebellum – showed impairments in the hidden (requiring the use of spatial strategies) but not the visually-cued condition of the Morris Water Maze task (Goodlett et al., 1992). Finally, unilateral removal of a cerebellar hemisphere impacts the spatial aspects in the Morris Water Maze task and inability to adapt behaviour to novel situations (Colombel et al., 2004; Petrosini et al., 1996; Leggio et al., 1999; Joyal et al., 1996). These pieces of evidence highlight that this connection might be important for spatial representations and the use of spatial strategies. This might be lateralised, expanding support for our findings that this organised topography might be involved in supporting specific aspects of behaviour. Further, there is growing evidence that the cerebellum directly influences hippocampal spatial representations, with disruption of cerebellar cells leading to alterations in hippocampal place cells firing. (e.g., Rochefort et al., 2013).”

2.3 Comment: *Further, the discussion of anterior-posterior hippocampal connectivity could be expanded to integrate functional distinctions of the cerebellar regions identified. For example, are anterior hippocampal connections more associated with cognitive functions beyond navigation, such as emotional regulation?*

2.3 Response: We thank the reviewer for this suggestion. We currently have a section relating these findings to functional distinctions within the cerebellum, but have extended this discussion as follows (shown in bold) (lines 231-249):

“Previous fMRI studies have implicated lobule HVIIA (Crus II) in non-motor functions such as first- and second-order rule learning (Balsters et al., 2013; Balsters and Ramnani, 2011), creative thinking (Gao et al., 2020), social/emotional processing (Guell and Schmahmann, 2020; Van Overwalle et al., 2020) and autobiographical memory (Addis et al., 2016). This is also consistent with its closer functional relationship with a “core” posteromedial cortical network (often termed the ‘default mode network’; Bucker et al., 2011), which – alongside the anterior hippocampus (Zeidman and Maguire, 2016) – is thought to support a range of constructive cognitive processes, including scene construction, autobiographical memory, and theory of mind (Spreng et al., 2009; Hassabis and Maguire, 2009). In contrast, a region more strongly associated with posterior hippocampus (lobule V) is integral to the motor control system. It is interconnected with cortical motor areas (Kelly and Strick, 2003), receives limb proprioceptive afference via the spinocerebellar system (see Oscarson, 1973), and contains fine-grained digit representations (Grodd et al., 2001; van der Zwaag et al., 2013).

This differential connectivity of anterior and posterior hippocampus to functionally distinct regions of the cerebellum suggests that functional subdivisions within the hippocampal formation – which are considered to emerge partially from differences in neocortical connectivity (Adnan et al., 2015; Aggleton, 2012; Dalton et al., 2019) – are also mirrored in cerebellum. The cerebellum is thought to form both motor and non-motor loops with different cortical areas (Ramnani et al., 2006). Our data suggest that the anterior hippocampus may interact preferentially with the cerebellar non-motor loop (consistent with its increased connectivity with prefrontal cortex; Cavada, 2000), whilst the posterior hippocampus may interact preferentially with the cerebellar motor loop. This distinction also aligns with current views of long-axis specialisation in the hippocampus, in which anterior hippocampus provides more abstract or coarse-grained representations in spatial and episodic memory, whereas the posterior hippocampus supports more fine-grained spatial processing (Poppenk et al., 2013; Robin and Moscovitch, 2017).”

2.4 Comment: *Although the study provides strong functional connectivity data, it does not confirm the presence of direct anatomical pathways. Incorporating diffusion MRI evidence (if available) and/or discussing potential relay stations (e.g., thalamic or brainstem structures) more explicitly would improve the anatomical grounding of the findings.*

2.4 Response: We thank the reviewer for this important point and fully agree that considering anatomical connectivity would be important for interpreting our functional connectivity results. In line with the reviewer's suggestion, we already provide a detailed discussion of potential relay stations – and the challenges associated with diffusion MRI in this context – on p. 6.

Animal studies remain inconclusive regarding the existence of a direct anatomical pathway between the hippocampus and cerebellum. Furthermore, there is limited consensus – aside from studies such as Watson et al. (2019) – on which regions may act as relay stations between these structures. To ensure that diffusion MRI tractography outputs are both informative and anatomically plausible, it is essential to identify appropriate anatomical waypoints to constrain tracking and reduce false positives. Candidate relay structures include the pontine nuclei, which receive input from the cerebral cortex and project to the cerebellum. However, the fibre architecture within the midbrain is highly complex and requires advanced tractography algorithms (e.g., constrained spherical deconvolution) to resolve reliably.

The cerebellum itself presents several additional challenges compared to the major fibre bundles of the cerebral cortex (see Lundell & Steele, 2024, for a detailed discussion). First, its small and highly convoluted folial organisation necessitates much higher spatial and angular resolution to accurately capture fibre orientations and reduce partial volume effects, which require ultra-high field/gradient MRI data that is not available in the dataset used here. The highly complex and densely crossed fibre geometries in the pontine nuclei (the main relay station for layer 5 output neurons from the cerebral cortex) also mean that trajectories are difficult to resolve. These difficulties are further compounded by the cerebellum's anatomical position, which can exacerbate image distortions and eddy current-related signal loss.

We have now extended our discussion to capture these points (lines 315-323):

'It may be possible to map structural pathways between cerebellar regions/nuclei and the human hippocampus by leveraging high-resolution diffusion MRI tractography. However, the fine-grained folial complexity of the cerebellum, coupled with the range and complexity of fiber populations (e.g., through the midbrain), would require advanced MRI acquisition and modelling approaches (Lundell and Steele, 2024), and potential validation data in other species. One study, for example, used probabilistic constrained spherical deconvolution tractography (which enables complex/crossing fiber populations to be better resolved) and identified tractography streamlines between the hippocampus and cerebellar lobules HVIII, IX, X, Crus I, Crus II and the fastigial nucleus (Arrigo et al., 2014), though it will be critical to validate these findings using ultra-high field/gradient diffusion MRI.'

2.5 Comment: *While motion correction and denoising strategies were implemented, residual age-related differences in head motion could contribute to connectivity reductions. A clearer discussion or additional control analyses addressing this issue would be beneficial. The authors should consider including head motion metrics (e.g., mean framewise displacement) as covariates in their statistical models to control for potential motion-related artifacts. Other relevant control variables that authors could include are cardiovascular health markers (e.g., blood pressure, BMI), cognitive performance scores, and white matter integrity metrics from*

diffusion MRI, which could better account for broader age-related changes that might influence functional connectivity.

2.5 Response: We thank the reviewer for raising this important point regarding motion-related artifacts. Our analysis applied the standard and widely recommended CONN (vR2020a) denoising pipeline. Specifically, it regresses six standard realignment parameters (three translational and three rotational) along with ART-based scrubbing regressors. Importantly, these are applied on a scan-to-scan basis, allowing for the removal of motion effect in the BOLD signal for each participant before connectivity is estimated – reducing its confounding influence on functional connectivity. This approach is well-validated and recommended (Whitfield-Gabrieli and Nieto-Castanon, 2012).

While we acknowledge the reviewers' suggestion to include mean framewise displacement as an additional second-level covariate, we note that the current preprocessing pipeline has been designed and implemented to minimise the effect of motion. Nonetheless, we appreciate that residual motion (following correction) may still influence our connectivity metrics/patterns, and we welcome further discussions on this.

In line with Reviewer 1's suggestion, we have also included BMI (mean-centred) and sex as covariates in our ageing models, and these are now reported in the Supplementary Materials. As noted in Response 1.3, the inclusion of these additional covariates had minimal impact on our results.

Regarding white matter integrity metrics, we refer the reviewer to Response 2.4, which outlines the significant methodological and conceptual challenges of tractography from cerebellar subregions.

As detailed in Response 2.2, behavioural data were not included due to the lack of relevant cognitive measures (i.e., spatial–mnemonic tasks) in this cohort. Furthermore, our study focuses specifically on the topography of hippocampal–cerebellar functional connectivity and its changes with age, rather than behavioural outcomes *per se*.

2.6 Comment: *The study uses a strict family-wise error correction, but some clusters are small (e.g., posterior hippocampal age effects). Reporting effect sizes or confidence intervals would help clarify the robustness of these findings.*

2.6 Response: We appreciate the reviewer's comment regarding the interpretation of small clusters, such as those observed in the posterior hippocampus in relation to age. We fully agree with the reviewer that effect sizes and confidence intervals can provide additional clarity regarding the robustness of results.

In this study, we applied strict family-wise error correction at the whole-brain level. As such, any cluster that survived this correction – regardless of the size – reflects a statistically robust effect.

Regarding effect sizes, the seed-based connectivity values from CONN are expressed as Fishers z-transformed correlation coefficients, which serve as effect size estimates. We believe that these provide a meaningful indication of the strength of the observed functional correlations. Nonetheless, we appreciate the suggestion of the reviewer, and are happy to provide further clarification, if required.

2.6 Comment: *The study primarily relies on the Harvard-Oxford and Jülich histological atlases for hippocampal delineation and the SUIT atlas for cerebellar mapping. While these are widely used, alternative atlases such as the Automated Anatomical Labeling (AAL) or the Schaefer 400 parcellation for functional connectivity may provide additional insights. For a*

finer-grained examination of hippocampal subfields, the FreeSurfer-based segmentation (e.g., the ex vivo hippocampal subfield atlas) could be utilized. Repeating analyses with one/some alternative atlases may refine the observed connectivity patterns and ensure robustness across different parcellation schemes.

2.6 Response: We thank the reviewer for these thoughtful suggestions. Our study was designed to conduct a focused investigation of hippocampal–cerebellar functional connectivity and its topographical organisation. For this reason, we used the SUIT atlas, which offers detailed and anatomically precise lobular cerebellar parcellation based on probabilistic estimates. While functional atlases such as the Schaefer 400 or AAL parcellation may be informative for broader cortical connectivity analyses, they do not provide additional insight into intra-cerebellar topography and therefore fall outside the scope of the present work.

With regard to hippocampal segmentation, we agree that finer-grained analyses using hippocampal subfields would be an important extension of this work. However, our resting-state fMRI data were acquired at a resolution of $3 \times 3 \times 4.44$ mm and further smoothed with a 5 mm full-width half-maximum kernel. This spatial resolution does not allow hippocampal subfields to be functionally delineated, which typically requires functional resolutions at or below 1.5 mm isotropic (see, e.g., Dalton et al., 2018; Hodgetts et al., 2017).

Moreover, although FreeSurfer-based hippocampal subfield segmentation from 1 mm T1-weighted images is widely used, such segmentations likely lack anatomical validity. In particular, T1-weighted images at this resolution do not allow visualisation of key internal landmarks, such as the stratum radiatum lacunosum moleculare (SRLM), which are essential for accurately distinguishing CA subfields from the dentate gyrus (see Wisse et al., 2020 for a detailed discussion of this issue).

In summary, our choice of atlases and parcellation schemes (e.g., anterior vs posterior hippocampus) reflects both the spatial resolution of our data and the anatomical precision required for our research question. We share the reviewer's interest in exploring hippocampal subfields in future work.